# Porous Tantalum vs. Titanium Implants: Enhanced Mineralized Matrix Formation after Stem Cells Proliferation and Differentiation

**DOI:** 10.3390/jcm9113657

**Published:** 2020-11-13

**Authors:** Sofia Piglionico, Julie Bousquet, Naveen Fatima, Matthieu Renaud, Pierre-Yves Collart-Dutilleul, Philippe Bousquet

**Affiliations:** 1Laboratory Bioengineering Nanosciences LBN, University of Montpellier, 34193 Montpellier, France; sofia-silvia.piglionico@etu.umontpellier.fr (S.P.); julie.bousquet@etu.umontpellier.fr (J.B.); naveen.fatima@etu.umontpellier.fr (N.F.); matt.renaud18@live.fr (M.R.); philippe.bousquet@umontpellier.fr (P.B.); 2Faculty of Dentistry, National University of Cuyo, Mendoza M5500, Argentina; 3CSERD, CHU de Montpellier, 34193 Montpellier, France

**Keywords:** osseointegration, porous tantalum, dental implant, mineralized matrix, osteodifferentiation, dental pulp stem cells, biomaterials

## Abstract

Titanium dental implants are used routinely, with surgical procedure, to replace missing teeth. Even though they lead to satisfactory results, novel developments with implant materials can still improve implant treatment outcomes. The aim of this study was to investigate the efficiency of porous tantalum (Ta) dental implants for osseointegration, in comparison to classical titanium (Ti). Mesenchymal stem cells from the dental pulp (DPSC) were incubated on Ta, smooth titanium (STi), and rough titanium (RTi) to assess their adhesion, proliferation, osteodifferentiation, and mineralized matrix production. Cell proliferation was measured at 4 h, 24 h, 48 h with MTT test. Early osteogenic differentiation was followed after 4, 8, 12 days by alkaline phosphatase (ALP) quantification. Cells organization and matrix microstructure were studied with scanning electron microscopy (SEM) and energy dispersive X-ray spectroscopy (EDX). Collagen production and matrix mineralization were evaluated by immunostaining and histological staining. MTT test showed significantly higher proliferation of DPSC on Ta at 24 h and 48 h. However, APL quantification after 8 and 12 days was significantly lower for Ta, revealing a delayed differentiation, where cells were proliferating the more. After 3 weeks, collagen immunostaining showed an efficient production of collagen on all samples. However, Red Alizarin staining clearly revealed a higher calcification on Ta. The overall results tend to demonstrate that DPSC differentiation is delayed on Ta surface, due to a longer proliferation period until cells cover the 3D porous Ta structure. However, after 3 weeks, a more abundant mineralized matrix is produced on and inside Ta implants. Cell populations on porous Ta proliferate greater and faster, leading to the production of more calcium phosphate deposits than cells on roughened and smooth titanium surfaces, revealing a potential enhanced capacity for osseointegration.

## 1. Introduction

Oral diseases are widely prevalent around the world and all can lead to tooth loss. Data from the World Health Organization reports on the impact of these afflictions: worldwide, nearly 100% of adults have dental caries, 15 to 20% have severe periodontal disease, and close to 30% of those over 65 years old are completely edentulous [1]. Complications with teeth loss are due to several etiologies, and are impactful at the oral level (such as reduction of masticatory efficiency, altered neuronal/physiological sensation, alveolar bone remodeling, changes on microflora composition, reduction of mechanical chewing function), in addition to causing esthetic changes that can affect patients’ psychological well-being. Dental implants are now able to restore functionality in both partially and completely edentulous patients. Dental implants provide such a successful outcome and more than 1000 varieties are on the market. There are a variety of different designs, dimensions, and surfaces (i.e., topography, chemistry, and wettability) [2,3].

Among dental implant options, titanium implants have unique characteristics that have made this type of implant the primary choice among clinicians. These characteristics include biocompatibility, good mechanical properties, and osseointegration [4]. Biocompatibility is due to the formation of a stable oxide layer of titanium dioxide on its surface [5]. Osseointegration has been previously defined as the close contact between bone and an implanted material [6]. Improving this close bone-implant contact is of great importance, because primary implant failure is strongly related with poor osseointegration, with a prevalence of around 2% in the first few months after implantation [7,8]. Therefore, specific attention is dedicated to dental implant surfaces in order to improve osseointegration, thus, avoiding early failure. The first and most validated strategy adopted to improve bone-implant interface is the modification of surface topography by increasing its roughness [9,10]. This approach is based on the knowledge that microroughness of the surface modulates not only of platelet activity, but also of cellular behavior, increasing cell attachment, proliferation, and differentiation in osteogenic cells [11,12,13], by up-regulating the synthesis of specific mRNA corresponding to osteogenic markers [14,15].

To achieve surfaces with different degrees of roughness, manufacturers have developed two main processes: additive and subtractive processes. Additive processes refer to surface coatings, such as plasma-sprayed titanium, or calcium phosphate and hydroxyapatite coatings, but it also comprises processes such as oxidation, ion deposition, and sintering of metallic powder. Subtractive processes include acid-etching, electro and mechanical polishing, sandblasting micro-texturing (MTX) [12,16,17]. Despite all improvements, the use of titanium is not without concerns, especially related to its mechanical properties (high elastic modulus, low shear strength, frictional characteristics, corrosion [18], and immune reactions such as allergies [19]). Thus, alternative implant materials have been proposed. Among them, porous tantalum has long been used in orthopedics to enhance neovascularization, wound healing, and osteogenesis, and presents a great potential to replace or complement dental implants due to its biocompatibility, mechanical, and anticorrosive properties [20,21]. An important advantage of tantalum over titanium is its elastic modulus: porous tantalum modulus is around 2.5–3.9 GPa (close to cancellous bone 0.1–0.5 GPa and cortical bone 12–18 GPa), while titanium (106–115 GPa) and other metals used with implantation purposes (cobalt chromium and stainless steel, 210–230 GPa) are fairly higher [22]. Furthermore, tantalum is fabricated as a three-dimensional porous material with a high density of pores (~75–85% porosity), which makes it similar to trabecular bone [22]. This structure serves as a matrix for blood clot formation, containing growth factors that attract undifferentiated/progenitor cells from the surrounding environment.

Bulk or porous structures induce a significant difference of surface areas accessible for cell colonization. In the work presented here, we aimed to give an insight of the overall process occurring during cell colonization and osteodifferentiation. We considered the whole process of cellular colonization, as it would occur in clinical situation. We made a comparison between bulk titanium and porous tantalum, taking in account the dental implants already available on the market. Even though promising investigations have been conducted on porous titanium, to date, they have not led to clinical applications in dental practice [23,24].

Following the concept of cell recruitment and differentiation, we hypothesized that populations of progenitor cells could repopulate tantalum surfaces, enter, and proliferate into their pores, and produce a mineralized matrix as expected for complete osseointegration. To test our hypotheses, we considered the use of dental stem cells. We used dental pulp stem cells (DPSC) as a model of progenitor cells, as they are the most common source of dental stem cells, due to the ease of obtainment after the extraction of healthy wisdom teeth, their high amount in dental pulp, and their high rate of proliferation [25,26,27]. In order to analyze the influence of implant surfaces on cell adhesion, proliferation, and differentiation in osteogenic cells, we assessed DPSC behavior on various specimens of smooth titanium (STi; machined surface), rough titanium (RTi; Microtextured-MTX), and porous tantalum (Ta).

## 2. Experimental Section

### 2.1. Implant Materials

Three different types of metallic discs (diameter 1.4 cm and 3.3 cm, thickness 2 mm) were investigated: Ta, smooth titanium (STi), and rough titanium (RTi) (Zimmer Biomet, Palm Beach Garden, Florida, USA). Ta discs were prepared with an average pore diameter from 350 to 450 μm, with a porosity from 75% to 85%. STi discs were machined to have smooth surfaces. RTi made reference to MTX Ti discs that were treated with grit-blasting to obtain rough surfaces. All discs were sterilized by gamma sterilization.

### 2.2. Culture of Human DPSC

Pulp cells were recovered from extracted third molars, as previously described [28]. Informed consent was obtained from patients, in accordance with local ethical committee, and cells were stored in an authorized biological collection. After tooth surface cleaning with 2% chlorhexidine, the tooth crown was cut for removal of the pulp tissue. Soft tissue was digested for 1 h at 37 °C in a solution of 3 mg/mL collagenase and 4 mg/mL dispase (BD Biosciences, Bedford, MA, USA). After filtration through a 70 Falcon strainer, the solution was mixed with αMEM supplemented with 10% fetal bovine serum, 100 U/mL penicillin, and 100 µg/mL streptomycin (Invitrogen, Carlsbad, NM, USA). Cells were rinsed after 24 h to remove non-adherent cells, and incubated in culture flasks for 1 week at 37° under 5% CO_2_ atmosphere. Each disc was seeded with 80.000 DPSC in αMEM at 37 °C and 5% CO_2_ for 1 week. The medium was changed twice per week. A week after seeding, osteogenic supplements were incorporated into the medium to induce differentiation of DPSC into osteoblasts. They were L-ascorbate phosphate, monopotassium phosphate, β-glycerophosphate, and dexamethasone. The culture time varied according to the analyses performed. In order to know the composition and microstructure of the samples, cells were kept in culture with osteogenic medium for 3 weeks. Scanning electron microscopy and microanalysis by dispersion of X-ray energy (scanning electron microscopy (SEM)/energy dispersive X-ray spectroscopy (EDX)) were used for this purpose. To assess osteogenic differentiation, collagen production (immunostaining) and early mineralized matrix production (Red Alizarin) were also analyzed 3 weeks after seeding. Cell proliferation was measured at 4 h, 24 h and 48 h (MTT test), and early osteogenic differentiation through quantification of alkaline phosphatase (ALP) at 4, 8 and 12 days. All experiments were conducted in triplicate.

### 2.3. Proliferation Assays

We used MTT colorimetric test to assess cell viability and proliferation (enzymatic activity). This test is based on the reduction of the yellow tetrazolium ring by the mitochondrial succinate dehydrogenase of living cells, forming a violet precipitate in mitochondria. In these experiments, 500 µL of MTT solution at a concentration of 1 mg in PBS was used per well, and evaluated after 3 h by adding 200 μL of isopropanol. Absorbance values were recorded with a plate reader at 540 nm (ELX 800, BioTek, Winooski, VT, USA).

### 2.4. Scanning Electron Microscopy (SEM) and Energy Dispersion X-ray Spectroscopy (EDX)

Cell adhesion was evaluated after 24 h by SEM, to control efficient cell attachment onto the surfaces: cells were seeded at low density (10^3^ cells/mL) and set in αMEM at 37 °C and 5% CO_2_ for 24 h. Samples were then fixed in 2.5% glutaraldehyde for 1 h at room temperature, rinsed in PBS, dehydrated with ethanol in increasing concentrations, before chemical drying at a critical point in hexamethyldisilazane (HDMS). Images of singles cells were taken with SEM to control correct cell attachment.

Composition and microstructure of the samples were evaluated by SEM and EDX spectroscopy. After 3 weeks of incubation in osteogenic culture medium, samples were fixed in 2.5% glutaraldehyde for 1 h at room temperature. They were then treated as described above, before imaging with SEM and composition analysis by EDX spectroscopy.

All samples were imaged and analyzed with a FEI Helios SEM (10 kV accelerating voltage, under a pressure of 1.3 × 10^−3^ Pa), without metallization.

### 2.5. Cell Differentiation

We used ALP as a specific marker of early osteogenic differentiation and assessed its expression after 4, 8 and 12 days of DPSC differentiation. DPSC seeded on discs were incubated in osteogenic medium at 37 °C in 5% CO_2_. The colorimetric method used was based on the reaction of the p-nitrophenyl phosphate (pNPP) at pH 10.4 (Liquid Substrate System, SIGMA Aldrich, St. Louis, MO, USA) with ALP enzyme. This solution combines pNPP, buffer, and the required magnesium cations. Yellow products resulting from the reaction with ALP were quantified with a plate reader at 405 nm. Experiments were conducted in triplicate.

### 2.6. Extracellular Matrix Formation

Bone-like extracellular matrix was assessed by quantification of Type 1 collagen with immunostaining. After 3 weeks in an osteogenic culture medium, cells were fixed with 4% paraformaldehyde for 15 min at room temperature, and washed 3 times with PBS. Cells were permeabilized with Triton x-100 at 0.5% for 15 min, and non-specific antigens were blocked with 1% bovine serum albumin. Anti-collagen antibodies were incubated overnight at 4 °C at a dilution 1:100 in 1% BS1/PBS. Cell nuclei were stained by adding 1 µg/mL 4’,6-diamidino-2-phénylindole (DAPI) for 30 min. Samples were observed under fluorescence microscopy at an excitation wavelength of 360 nm for DAPI staining (nucleus staining) and 630 nm for anti-collagen antibodies. A minimum of 6 images were taken per sample (5 on the side, 1 at the center). Images were processed and analyzed by ImageJ software. All experiments were conducted at least in triplicate. Results were expressed as percentage of total surface covered by mature collagen (mean percentage ± standard deviation).

### 2.7. Calcium Deposits

To assess calcium deposits on the discs, we used Alizarin Red coloration after 3 weeks of culture in osteogenic medium. It is a colorant that binds to calcium, leaving a red staining after washing. After an incubation period, cells were washed in PBS and fixed with 95% ethanol for 30 min at room temperature. Samples were then washed again and incubated with red alizarin solution at 2% for 30 s. Excessive colorant was removed by washing 8 times in distilled water. Colorant from stained samples was detached from calcium particles with 10% (*w/v*) cetylpyridinium chloride for 15 min under gentle agitation. The colored liquids were collected and transferred into a 96 well plates. The calcium concentration was determined by absorbance measurements at 562 nm with a plate reader. Experiments were conducted in triplicate, with results expressed in arbitrary unit (optical density—OD) as mean ± standard deviation.

### 2.8. Statistical Analysis

All data were evaluated with the Shapiro–Wilk normality test. Results were plotted as mean ± standard error of the mean and statistical analyses were performed using an ANOVA test (SigmaStat, 356 SPSS Inc., Chicago, IL, USA) or a Mann–Whitney non-parametric test, if the data did not pass the normality test. A *p*-value of <0.05 was considered to be significant.

## 3. Results

### 3.1. Cell Adhesion and Proliferation

After 24 h, all observed DPSC were properly attached on the various surfaces, with elongated shape, and cell protrusions revealing anchorage points. Representative images of DPSC on Ta and Ti are shown in Figure 1. For cell proliferation from 4 h to 48 h, the MTT assays showed a higher DPSC proliferation rate on Ta compared to both specimens of Ti alloys (Figure 1). Data were normally distributed, and an ANOVA test revealed that the differences observed between Ta and Ti were statistically significant after 24 h and 48 h (*p* = 0.005 and *p* < 0.001 respectively).

### 3.2. Cell Osteodifferentiation

Osteodifferentiation during the 2 first weeks of incubation in osteogenic medium were assessed by ALP quantification with a colorimetric assay. After 4 days, optical density results obtained from the 3 groups did not show any difference (*p* = 0,180) (OD_Ta_ = 0.11, OD_RTi_ = 0.09, OD_STi_ = 0.10). After 8 days, some differences could be observed, with a higher osteodifferentiation into Ti surfaces, but without statistical significance (*p* = 0.140) (OD_Ta_ = 0.26, OD_RTi_ = 0.3, OD_STi_ = 0.39). After 12 days differences increased even more, showing that osteodifferentiation was statistically higher on both Ti surfaces compared to tantalum surface (OD_Ta_ = 0.37, OD_RTi_ = 0.69, OD_STi_ = 0.55; *p* < 0,001). All data were normally distributed, and analyzed with ANOVA test. ALP quantification results are summarized in Figure 1.

### 3.3. SEM and EDX

After 3 weeks of culture in osteogenic medium, samples were observed with SEM, and the following atomic elements were quantified (in percentage) with EDX spectroscopy: carbon, oxygen, phosphorus, calcium, titanium, tantalum. The quantification of these elements allowed to consider the Ca/P ratio, in order to give insight of the mineral structures. Indeed, the Ca/P ratio and Ca/O ratio can be used as reference to define the various Calcium–Phosphate crystals found in biological systems (Table 1). In our experiments, the high amount of oxygen detected from cellular structures did not allow to use the Ca/O ratio, as most O was coming from the cells covering the samples. The observed percentages varied according to the location, leading to data distribution that did not follow normal distribution. Thus, Ca/P ratio were compared using the Mann–Whitney non-parametric test.

On RTi surfaces, discs were densely covered by cells appearing as black strips with white dots and small plates surrounding them (Figure 2A,B). The composition of calcium and phosphate varied from 1 to 7%, with a Ca/P ratio of 1.27 ± 0.08, placing the mineral structures close to octacalcium phosphate crystals.

The STi surfaces were also fully covered by cells, with a particular disposition, following the spiral-like pattern of the surface, linked to sample preparation (cutting and polishing) (Figure 2C,D). The composition of calcium and phosphate varied from 1 to 5%, with a Ca/P ratio of 1.21 ± 0.12, placing the mineral structures between dicalcium phosphate and octacalcium phosphate crystals.

The porous structure of Ta samples was clearly visible with SEM, presenting a structure of adjoining cavities, similar to honeycomb, with cells covering the edge and filling partially the cavities (Figure 2E,F). Tantalum appeared bright under SEM because of its inorganic composition, while cells were darker due to their organic composition. Trapped between cellular nets white cores could be observed, composed of Ca and P (as confirmed by EDX spectroscopy). The composition of calcium and phosphate varied from 1.5 to 4%, with a Ca/P ratio of 1.28 ± 0.11, corresponding to mineral structures close to octacalcium phosphate crystals, with some areas close to hydroxyapatite. Results from EDX spectroscopy analyses are summarized in Table 2 and Figure 2.

### 3.4. Collagen Immunostaining

To assess osteogenic differentiation, we quantified by immunostaining the production of type I collagen, which is found in mature extracellular matrix. After 21 days of culture, DPSC differentiation was determined for each sample by the expression of type I collagen, after immunostaining with anti-collagen antibodies, making it possible to determine the percentage of the surface covered by mature collagen. Cell nuclei were counterstained in blue. We considered the whole surface covered by collagen for analysis, expressing it as percentage of the total surface (Figure 3). All data were normally distributed, and a comparison was performed with ANOVA test.

Overlaid images showed the RTi samples almost completely covered by collagen with a high quantity of nucleus. The analysis of surface coverage revealed that 79.5 ± 11.2% of the samples were covered with mature (Figure 3A).

In STi specimens, overlaid images are similar to those of RTi. They showed an even more dense distribution of collagen in red and a large presence of nucleus (Figure 3B). The analysis of the area showed important number of cells (stained nuclei) with the area occupied by collagen representing 82.9 ± 12.5% (Figure 3).

Ta discs were densely covered by collagen, even though nuclei seemed sparse (Figure 3C). The mean area covered by collagen was 81.7 ± 7.8%. This percentage was measured by taking into account only the surface available for observation in the focal plan, thus, excluding the holes due to porous structure (areas delimited by dash lines in Figure 3C).

### 3.5. Calcium Deposits Quantification

After 21 days of osteodifferentiation, calcium deposits were stained with Red Alizarin. Samples could not be observed directly under optic microscope due to their opacity. Analyses were conducted indirectly by quantifying red alizarin release after treatment with cetylpyridinium chloride. Calcium concentrations were determined by measuring absorbance at 562 nm with a plate reader. Thus, we did not quantify only calcium visible in focal plan, as for collagen immunostaining, but the whole amount of calcium on the surface and inside the pores. Data were normally distributed. Results showed a significantly higher quantity of calcium on Ta discs compared to Ti specimens with ANOVA test (OD_RTi_ = 0.69 ± 0.11, OD_STi_ = 0.88 ± 0.21, OD_Ta_ = 2.8 ± 0.22, *p* < 0.01). These results are summarized in the histogram in Figure 3.

## 4. Discussion

Implant surfaces evaluated during this research have already been assessed and compared in different conditions by others groups of researchers, who mostly considered Ta as a good alternative to Ti surfaces implants, not only because of its good mechanical properties, but also because it has proven to be favorable for cell adhesion, proliferation, osteodifferentiation, and mineralization in vitro and in vivo [29,30]. In this study, we focused on the progressive process of cell invasion and bone matrix formation, by mimicking in vitro the clinical situation on dental implant graft: progenitor cells colonize the biomaterial surface, proliferate, then differentiate to produce a mineralized matrix surrounding the implanted device. DPSC were used as a model of mesenchymal progenitor cells.

As already described in previous studies, both Ti alloys and Ta were efficient for cell adhesion and initial proliferation, as shown by MTT tests after 4 h and SEM observation after 24 h. Cell shape with several protrusions such as filopodia and lamellipodia were objective signs of proper cell attachment, with cells starting to colonize the surfaces [29]. Furthermore, SEM images taken after 3 weeks revealed that samples were fully covered by cells, confirming high cell proliferation and coverage of the surfaces in all the groups (Ta, STi, RTi). However, MTT tests allowed to compare early proliferation on the various samples and revealed a significantly greater DPSC proliferation on Ta discs compared to Ti specimens after 4 h, 24 h and 48 h. These results are in accordance with a previous study using osteoblastic lineage on Ti and Ta samples, with an observation of increased cell proliferation for the Ta group [31]. There was a clear plateau for DPSC proliferation on RTi and STi beyond 48 h, whereas cells continued to proliferate on Ta (MTT assay, Figure 1). Indeed, the porous structure of Ta samples offered an increased surface area with subsequent enhancement in cell proliferation. The proliferation process, where cells replicate rapidly, is limited over time. Once undifferentiated cells become confluent, having close contact with each other, downregulation of DNA replication starts. At that stage, expression of early osteoblast markers such as ALP and collagenic proteins starts to be mildly perceptible. Results showed increasing values of ALP expression during the evaluation period for all the groups. However, lower levels of ALP production were quantified on Ta after 8 and 12 days, compared to both Ti specimens. These data differ from results obtained in a previously published study using osteoblastic lineage [32], where ALP measurements were rapidly increasing over time (1, 3, and 7 days), especially on tantalum substrate compared to titanium surfaces. However, these assays were conducted with preosteoblasts, a more differentiated lineage, while we were following undifferentiated cells that had to proliferate to reach full confluence before starting differentiation process. Another research work had described that Ta flat surfaces could significantly increase ALP levels for both undifferentiated and osteogenically stimulated mesenchymal stem cells, even after few days of seeding [33]. These previous works showed efficient cell differentiation on flat tantalum substrates for both progenitors (mesenchymal stem cells) and predifferentiated cells (preosteoblasts). In our experimentation, considering both MTT assays and ALP quantification, we could observe that undifferentiated DPSC were still proliferating on Ta after 48 h. This cellular behavior was consistent with a delayed differentiation, as objectivated by lower ALP production after 8 and 12 days. Thus, we could hypothesize that the porous structure of Ta samples led to longer cell multiplication until complete coverage/filling of the scaffold. Indeed, compared to flat surfaces, porous Ta has a greater surface area due to its porous structure. The differentiation process on porous Ta started later, resulting in increased total number of cells for differentiation and extracellular matrix section. This could then lead to more abundant and efficient mineralization (calcium deposits quantification, Figure 3).

In a classical osteodifferentiation process, the transformation into preosteoblast, then osteoblast can further be detectable by the production of bone marker proteins such as osteocalcin, osteonectin, osteopontin, and bone sialoprotein. The last phase of osteoblast development is defined by the nucleation of hydroxyapatite crystals along mature type 1 collagen fibers [34]. One decisive point for this late osteoblastic differentiation is the contact of cells with type 1 collagen matrix before starting the expression of osteoblastic genes [35,36]. We took into account the importance of this matrix and analyzed the type 1 collagen fibers produced after 3 weeks, on Ta, STi, and RTi, by collagen immunostaining. Results assessed a high type 1 collagen expression all over the surfaces, in all the groups, confirming the efficient cell coverage and osteodifferentiation, even without considering the 3D aspects of porous Ta.

Regarding the concept of crystal nucleation, Ta is theoretically a promising material, due to the formation of Ta-OH groups on its surface that can facilitate adsorption of calcium (Ca) and phosphate (P), improving osteoblasts adhesion, proliferation, and differentiation leading to successful osseointegration [37,38]. In accordance with a previous study where EDX plots showed high levels of Ca and P inside porous tantalum after 2 weeks of implantation [39], we also observed Ca and P deposits trapped into the cellular net formed in Ta pores. With EDX spectroscopy, these nodules were difficult to localize for their analysis, due to the 3D disposition of cellular tissue into the pores that cover these crystals as translucent blankets. However, we could observe the presence of nodules, which, according to Ca/P ratio, could be mostly related to octacalciumphosphate crystals. Calcium phosphate nodules were very similar on RTi and Ta, while there were more dicalcium phosphate nodules on STi (Figure 2 and Table 2).

With Red Alizarin staining, we were able to quantify the total amount of mineralization after 3 weeks. Thus, the large quantities of calcification products were clearly visible, with significantly more abundant calcium deposits for Ta groups. These data confirm that calcium phosphate crystals were produced along collagen fibers even inside Ta porosities, and are consistent with previous studies on porous Ta (using histological analysis) where bone ingrowth was observed within porous Ta implants during healing process with healthy patients [40], and even with osteopenic patients [41].

Tantalum is a rare transitional metal with high corrosion resistance properties. The rarity of Ta, and subsequent high manufacturing cost, had limited its applications in medical fields. However, porous Ta produced via chemical vapor deposition of commercially pure Ta onto a vitreous carbon is currently available for use in orthopedic applications, and porous tantalum trabecular metal (PTTM) has been used in orthopedic implants for several years [42]. Tantalum has recently been tried for incorporation into Ti or coating onto Ti alloy [38,43].

To date, Ti implants remain the reference for bone implants. To improve well-known titanium implants, porous structures have been designed for increasing bone formation and close bone-implant interface. Selective laser melting (SLM) or electron beam melting (EBM) are the main manufacturing techniques used to produce 3D porous Ti implants [44]. Pore structure mimicking human trabecular bone with interconnected porous network can induce cell ingrowth, migration, and differentiation [23]. Most in vitro and in vivo studies indicate that scaffolds with porous structures are superior for osteoblast growth, which could be attributed to larger surface area for cell penetration inside the porous architecture. Concerning cell differentiation, no obvious modification can be described when considering osteogenic markers expression for bulk or porous scaffolds.

To further compare our results (on porous Ta) to previous studies on porous Ti, we made a bibliographic analysis to gather all in vitro experimentations conducted on porous Ti scaffolds with similar porosity (pore size ranging from 200 to 1000 µm and/or porosity ranging from 60 to 80%). The main results of the 13 selected studies are presented in Table 3. Taken altogether, these compared results indicate that porous Ti retain their cytocompatibility (with either SLM or EBM manufacturing techniques), and that the porous architecture increase cell proliferation, with a tendency for small pore size (<500 µm) to yield higher cell proliferation. No obvious positive effect on cell osteodifferentiation could be described. Similarly, our results observed an increased cell proliferation, probably linked to the porous architecture, with no significant difference when considering cell differentiation. However, in the considered previous studies, a coating with Ta or hydroxyapatite improved osteogenic differentiation and bone growth. In the research work presented here, we confirm the positive impact of tantalum on bone-like matrix formation and mineralization.

A recent clinical study demonstrated that porous Ta dental implants induced an upregulation of specific genes implicated in neovascularization and osteogenesis, compared to Ti implants [42]. Our data confirm clinical studies that elucidate advantages of Ta implants. An uncontrolled cross section study observed that Ta dental implants were clinically efficient under diverse clinical conditions [55]. Further clinical research confirmed that Ta implants exhibited less peri-implant bone loss than Ti specimens [56]. Taken altogether, our results give an overview of the healing process of proliferation-differentiation-mineralization around titanium and porous tantalum implants. Osteodifferentiation is delayed with porous Ta implants, due to increased cell proliferation rate. After 3 weeks in vitro, a 3D complex matrix is formed inside Ta pores, completely colonized by osteogenic cells producing more abundant mineralized nodules along the collagenic extracellular matrix. The translation of this process in clinical practice would explain the observed more efficient osseointegration and stronger implant stability of Ta implants, when compared to conventional Ti implants. It reinforces the interest of porous implants fulfilling both aspects of biocompatibility (efficient cell colonization) and mechanical characteristics (low Young Modulus, closer to bone).

## 5. Conclusions

Porous Ta implants are already available for clinical use, with satisfying results, as for Ti implants. We present, in this work, rational explanations to objectivate pragmatic clinical observations. Tantalum porous structures give greater surface areas available for cell attachment and proliferation, when compared with conventional titanium surfaces, due to their porosity and structural irregularities. These structures allow cells to proliferate for a longer period, until completely covering/filling the 3D structures. Later on, the differentiated cell populations on porous Ta produce more calcium-phosphate crystals than cells on roughened or smooth titanium surfaces, leading to potential enhanced osseointegration.

## Figures and Tables

**Figure 1 jcm-09-03657-f001:**
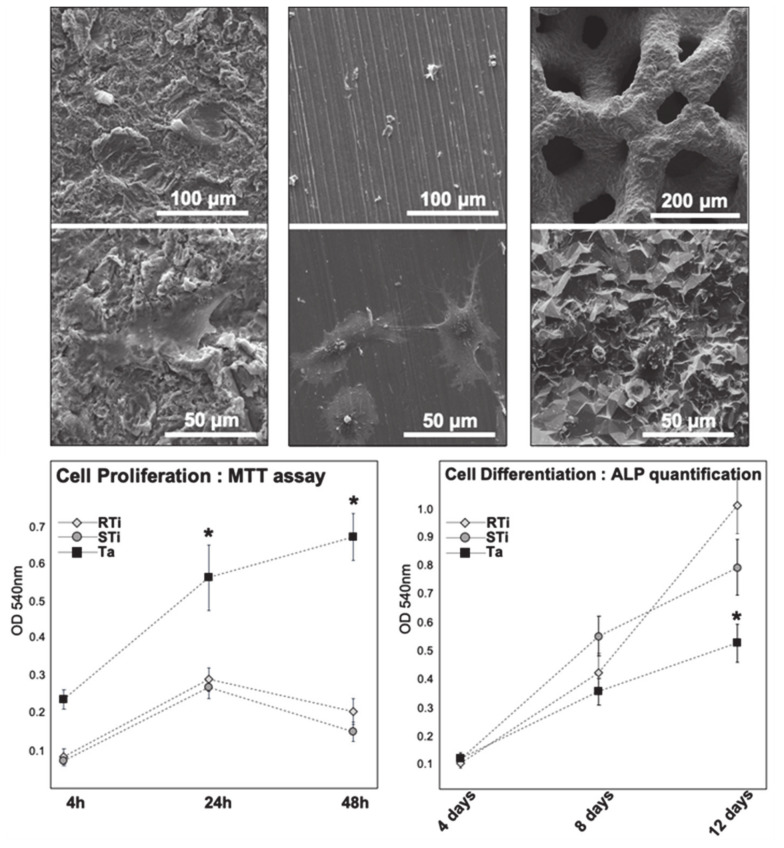
Scanning electron microscopy (SEM) images of dental pulp stem cells (DPSC) after 24 h of incubation, at a magnification ×400 and ×2000. From left to right: rough titanium (RTi), smooth titanium (STi), and tantalum (Ta). At higher magnification, spread and elongated cells are visible. Lower graphs: optical density from colorimetric assays (absorbance at 540 nm). Left graph: cell proliferation evaluation after 4 h, 24 h, and 48 h with MTT assays. Right graph: cell differentiation evaluation after 4, 8, and 12 days with alkaline phosphatase (ALP) quantification. * indicates significant difference (*p* < 0.05).

**Figure 2 jcm-09-03657-f002:**
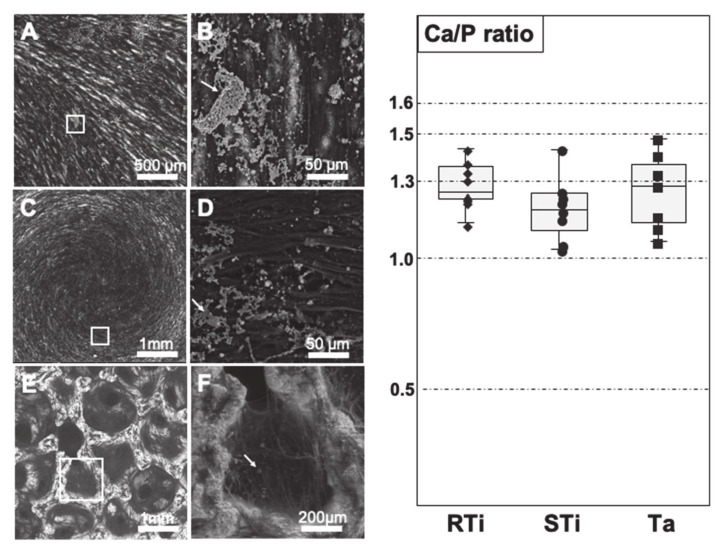
SEM images of DPSC after 3 weeks in osteogenic medium. (**A,B**) RTi at magnification ×100 and ×1500. (**C,D**) STi at magnification ×100 and ×1500. (**E,F**) Ta at magnification ×100 and ×400. White squares on lower magnification images indicate the area of magnification. White arrows indicate calcified nodules. Right graph: box plot representation of Ca/P ratio as calculated after energy dispersive X-ray (EDX) spectroscopy on the various samples.

**Figure 3 jcm-09-03657-f003:**
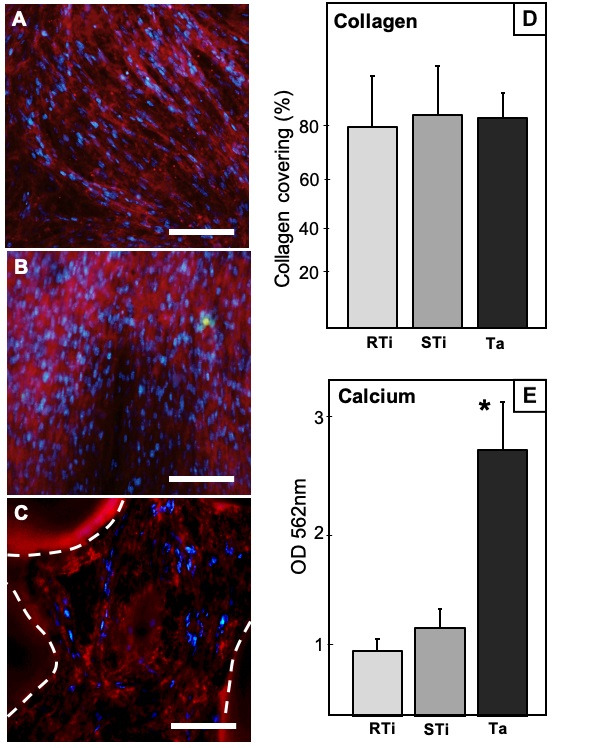
(**A**–**C**) Epifluorescence microscopy images of collagen immunostaining. RTi, STi, and Ta, respectively. Type 1 collagen fibers are stained in red. Cell nuclei are stained in blue. Dash lines on picture C indicate the area of Ta substrate in the focal plan. (**D**) Percentage of surface covered by stained collagen, as assessed by ImageJ analysis. (**E**) Calcium deposits quantification, after Red Alizarin staining. Optical density from colorimetric assays (absorbance at 562 nm). * indicates significant difference.

**Table 1 jcm-09-03657-t001:** Theoretical Ca/P and Ca/O ratio for various calcium phosphate crystals.

Crystal Structure	Ca/P Ratio	Ca/O Ratio
MonoCalcium Phosphate	0.5	0.125
DiCalcium Phosphate	1	0.25
OctaCalcium Phosphate	1.33	0.275
TriCalcium Phosphate	1.5	0.375
Hydroxyapatite	1.67	0.55

**Table 2 jcm-09-03657-t002:** Relative percentage (mean value) of atomic elements recovered by EDX spectroscopy.

% Atomic Element	R Ti	S Ti	Ta
C	21.4%	24.9%	19.7%
O	65.1%	65.2%	57.1%
P	2.7%	2.2%	2.0%
Ca	3.5%	2.7%	2.6%
Ti	4.2%	2.2%	-
Ta	-	-	17.0%
Ca/P Ratio	1.27	1.21	1.28

**Table 3 jcm-09-03657-t003:** Main results from previously published after *in vitro* studies on porous titanium scaffolds, with comparable pore size and/or porosity.

Reference	Pore Size Range	Porosity	Cell Type	Conclusion
Warnke 2009 [45]	400–1000 µm	70–80%	Osteoblast	Increased cell proliferation on scaffolds with 400 µm pore size
Van Bael 2012 [46]	500–1000 µm	70–80%	Periosteum Cell	Increased cell proliferation and cell osteodifferentiation on scaffolds with 500 µm pore size
Li 2013 [43]	400–600 µm	75–85%	BMMSC	Better cell adhesion and proliferation on porous scaffolds coated with tantalum. No difference of bone formation between coated and uncoated
Amin Yavari 2014 [47]	500 µm	88%	Periosteum Cell	Anodizing-heat treatment improves cell attachment and proliferation, and osteogenic markers expression.
Matena 2015 [44]	250 µm	50%	Osteoblast	Lower cell adhesion on PCL coated Ti scaffolds, but increased chemotactic behavior for endothelial cells
Markhoff 2015 [48]	500–1000 µm	50–70%	Osteoblast	SLM and EBM manufacturing techniques lead to similar cell adhesion, proliferation and osteodifferentiation
Wysocki 2016 [49]	200–500 µm	70%	MSC	Smaller pores improved cell adhesion and proliferation.
Wang 2016 [50]	1000 µm	75%	MSC	SLM and EBM porous Ti scaffolds are cytocompatible
Wang 2016 [38]	nd	30–50%	MG63 Cell	Ta-coated porous Ti implants improve cell adhesion, proliferation and osteodifferentiation (compared to uncoated)
Fousová 2017 [51]	1000 µm	60–80%	Osteoblast	SLM porous Ti implants retain Ti alloy cytocompatibility
Wang 2019 [52]	500 µm	70%	BMMSC	Both porous Ta and porous Ti scaffolds were in favor of BMMSC proliferation and osteogenic differentiation
Bartolomeu 2020 [53]	500–600 µm	64–93%	Fibroblast	Porous Ti scaffolds produced by SLM did not release toxic substances and insured a suitable environment for cell proliferation
Liu 2020 [54]	400–1000 µm	70–85%	Osteoblast	All porosities (porous Ti gradients) investigated were suitable for cell adhesion and cell survival

nd: not determined.

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
