# Peer review of "Porous Tantalum vs. Titanium Implants: Enhanced Mineralized Matrix Formation after Stem Cells Proliferation and Differentiation"

_jcm, 2020, doi:10.3390/jcm9113657_

Round 1
Reviewer 1 Report
Dear authors,
Thank you very much for this wonderfully written paper. It is well structured, the methods are sufficient and the topic is interesting.
I would have liked two things:
- A more precise description which data were normally distributed and which tests were used for each group.
- A few sentences about the practical consequences of your findings in the discussion. Is it advisable to research Ta implants or does this material even possibly offer better material characteristics than Ti?
Thank you.
Reviewer 2 Report
This work sounds interesting, however,
I have few comments and suggestions as following before further consideration,
1. For cell adhesion and proliferation results, Ta showed highest cell proliferation. However, Ta disks had porous structure with porosity 75% to 85% and pore diameters from 350 to 450 micro-meter whereas Ti only flat/rough surface. Porous structure provided more surface areas facilitated cell attachments and proliferation. Authors are recommended to consider this factor which affected the MTT results.
2. Authors are recommended to consider the comparison between porous Ta and porous Ti with similar porosity and pore diameters.
3. There are 5 figures in figure 3. The labels were only showed A, B, C.
4. page 7, line 49: there is no figure 4. Please correct it.
